# What Promotes Acute Kidney Injury in Patients with Myocardial Infarction and Multivessel Coronary Artery Disease—Contrast Media, Hydration Status or Something Else?

**DOI:** 10.3390/nu15010021

**Published:** 2022-12-21

**Authors:** Joanna Maksimczuk, Agata Galas, Paweł Krzesiński

**Affiliations:** Department of Cardiology and Internal Diseases, Military Institute of Medicine—National Research Institute, 128 Szaserów Street, 04-141 Warsaw, Poland

**Keywords:** acute kidney injury, myocardial infarction, multivessel coronary artery disease, hydration status

## Abstract

Multivessel coronary artery disease (MVCAD) is found in approximately 50% of patients with acute myocardial infarction (AMI) undergoing percutaneous coronary intervention (PCI). Although we have data showing the benefits of revascularization of significant non-culprit coronary lesions in patients with AMI, the optimal timing of angioplasty remains unclear. The most common reason for postponing subsequent percutaneous treatment is the fear of contrast-induced acute kidney injury (CI-AKI). Acute kidney injury (AKI) is common in patients with AMI undergoing PCI, and its etiology appears to be complex and incompletely understood. In this review, we discuss the definition, pathophysiology and risk factors of AKI in patients with AMI undergoing PCI. We present the impact of AKI on the course of hospitalization and distant prognosis of patients with AMI. Special attention was paid to the phenomenon of AKI in patients undergoing multivessel revascularization. We analyze the correlation between increased exposure to contrast medium (CM) and the risk of AKI in patients with AMI to provide information useful in the decision-making process about the optimal timing of revascularization of non-culprit lesions. In addition, we present diagnostic tools in the form of new biomarkers of AKI and discuss ways to prevent and mitigate the course of AKI.

## 1. Introduction

Multivessel coronary artery disease (MVCAD) is found in approximately 50% of patients with acute myocardial infarction (AMI) undergoing percutaneous coronary intervention (PCI) [1]. Patients diagnosed with MVCAD are often older, have multiple risk factors for atherosclerosis, have previously suffered from AMI and underwent PCI or coronary artery bypass grafting [1]. As shown in numerous studies, the presence of MVCAD is associated with worse short- and long-term outcomes [1,2]. The available data indicate that revascularization should not only address the lesion in the culprit artery, but should also include the remaining significant lesions in the coronary arteries, as patients undergoing complete revascularization have a lower risk of adverse cardiovascular events [3,4,5,6,7]. The current recommendations of the European Society of Cardiology (ESC) for the management of both ST-elevation myocardial infarction (STEMI, 2017) and non-ST-elevation myocardial infarction (NSTEMI, 2020) recommend class IIa multivessel revascularization (level of evidence A and C) [8,9]. The optimal timing for revascularization of non-culprit lesions remains unclear. The aforementioned ESC recommendations encourage complete revascularization before the patient is discharged from the hospital. Those for the management of STEMI indicate revascularization of non-culprit lesions during a single procedure or in stages (recommendation class IIa, level of evidence A). According to ESC recommendations regarding NSTEMI, complete revascularization in patients with MVCAD may be considered during a single procedure (recommendation class IIb, level of evidence C). Early revascularization of non-culprit arteries in patients with MVCAD may improve overall perfusion and myocardial function, guarantying better outcome of patients in the acute phase of AMI and shortening the length of hospitalization [10]. Nonetheless, in daily clinical practice, there may be concern about the use of early multivessel revascularization strategies due to the risk of worsening renal function as a consequence of applying a larger volume of iodinated contrast media (CM) in a short period of time.

## 2. Acute Kidney Injury in Patients with Myocardial Infarction

### 2.1. Definition, Epidemiology and Impact on Prognosis

In the past, various definitions of acute kidney injury (AKI) were proposed, which was the source of many inaccuracies. Now in widespread use are the 2012 standardized criteria developed by the Kidney Disease: Improving Global Outcomes (KDIGO) group. According to the KDIGO criteria, AKI is defined as an increase in serum creatinine of at least 0.3 mg/dL within 48 h or 1.5 times baseline, known or presumed to have occurred within the past 7 days, or based on urine output, a urine volume of less than 0.5 mL/kg/h for at least 6 h [11]. Other definitions used in studies include those according to RIFLE and AKIN [12,13] (Table 1).

Acute kidney injury (AKI) is common in patients with AMI undergoing PCI. The reported incidence of AKI in patients hospitalized for AMI varies from study to study, ranging from 10 to 30% [14,15,16,17,18,19]. These differences are due to the varying definition of AKI adopted in a given study and the different characteristics of the groups of participating patients. As shown in many studies, the occurrence of AKI in patients with AMI not only prolongs hospitalization and increases its costs, but is also associated with higher mortality and morbidity in both short- and long-term follow-up [15,20,21]. A retrospective analysis of clinical data by Wang et al. showed significantly higher in-hospital mortality in a group of patients with AMI who developed AKI during hospitalization, compared to a group of patients without a diagnosis of AKI (20% vs. 0.6%) [17]. Sun et al. showed a strong association of AKI occurrence with both short- and long-term mortality [22]. While in a study by Chalikias et al. mortality during the 3-year follow-up period among patients who developed episodes of AKI during hospitalization was three times higher compared to those without a diagnosis of AKI. Moreover, in that study, during long-term follow-up, patients who developed episodes of AKI during hospitalization had significantly more episodes of major adverse cardiovascular events (MACE) (26% vs. 15%) and worsening of renal function (10 vs. 5%) [14]. Even slight deterioration of renal function, without meeting the criteria for AKI, has been shown to be associated with a worse long-term outcomes in patients hospitalized for AMI [23].

### 2.2. Etiology, Pathogenesis

When talking about the etiology of renal injury in AMI patients undergoing PCI, the effect of CM on renal function is traditionally mentioned. However, the etiology of renal injury in patients with AMI is much more complex and results from multiple mechanisms acting synergistically [24,25,26]. Hemodynamic abnormalities resulting from reduced cardiac output and venous return consequently lead to a decrease in glomerular filtration rate (GFR). In addition, patients with AMI experience activation of the sympathetic nervous system and the renin-angiotensin-aldosterone system (RAAS), which can lead to vasoconstriction and exacerbate kidney damage [27,28]. Activation of the inflammatory response and oxidative stress also has deleterious effects on renal function by damaging renal tubular cells, and consequently reducing the ability to reabsorb sodium and water, decreasing circulating blood volume and thus further impairing renal perfusion [29]. Another possible element in the pathomechanism of renal damage in patients with AMI is disturbances in the coagulation system. Platelet activation can promote microthrombosis in the glomeruli, causing nephron damage and ultimately leading to deterioration of renal function [30]. In addition, metabolic factors such as hyperglycemia and acidosis have an adverse effect [31,32]. Other factors involved in the pathogenesis of AKI in patients with AMI include atherosclerotic embolism caused by the release of plaque fragments, bleeding, medications and intra-aortic counterpulsation.

### 2.3. Role of Contrast Agent

The risk of contrast-induced acute kidney injury (CI-AKI) is a topic of particular importance in AMI patients undergoing multivessel revascularization due to the use of larger volumes of CM. The mechanism of renal injury by CM is complex and includes direct and indirect effects. CM is directly toxic to renal tubular epithelial cells, leading to their apoptosis and necrosis. CM causes loss of cell polarity due to redistribution of Na^+^/K^+^-ATPase in the membrane of tubular epithelial cells, resulting in abnormal ion transport and increased sodium delivery to distal tubules. This phenomenon leads to renal vasoconstriction via tubular-glomerular feedback. As the damage progresses, epithelial cells detach from basement membranes, causing lumen obstruction, an increase in intra-tubular pressure, and, eventually, a decrease in GFR. Indirect effects of CM include ischemic damage caused by contraction of intrarenal arterioles due to an imbalance of vasoactive factors such as endothelin, nitric oxide and prostaglandin. The result is reduced blood flow in the glomeruli and reduced oxygen delivery to the kidney’s outer medulla. Renal vasoconstriction and renal ischemia also lead to the formation of tissue-damaging reactive oxygen species. In addition, CM increases blood viscosity, causing impaired microcirculation and changes in blood osmolality, which, in turn, impairs erythrocyte plasticity and may increase the risk of microvascular thrombosis [33] (Figure 1).

The incidence of CI-AKI in patients undergoing PCI ranges from 2 to 20% and is higher in patients undergoing urgent revascularization [34]. The 2012 KDIGO guidelines [11] and recent statements from the Italian College of Radiology (SIRM), the Italian College of Nephrology (SIN) and the Italian Association of Medical Oncology (AIOM) [35] mention pre-existing renal functional impairment as the most important risk factor for CI-AKI. According to KIDGO guidelines, the risk of CI-AKI becomes clinically significant when the GFR is <60 mL/min/1.73 m^2^. The SIRM-SIN-AIOM document cites the European Society of Urogenital Radiology (ESUR) guidelines and states the GFR value < 45 mL/min/1.73 m^2^ as significant for intra-arterial administrations with renal pass and <30/mL/min/1.73 m^2^ for intravenous and intra-arterial administrations with second pass. Diabetes is a questionable risk factor for CI-AKI, but it multiplies the risk when coexisting with chronic kidney disease. Other major risk factors for CI-AKI are: heart failure, hypovolemia, older age, anemia, myocardial infarction, peripheral vasculopathy and the use of nephrotoxic drugs. There are studies showing a correlation between the volume of CM used and the incidence of CI-AKI. In a study involving 561 STEMI patients treated with PCI, higher CM volume was associated with higher risk of AKI [36]. Furthermore, analysis of data from the HORIZONS-AMI trial showed that CM volume was an independent predictor of AKI incidence in STEMI patients undergoing PCI [37]. Another study involving 2308 patients undergoing PCI showed that the risk of CI-AKI was significantly higher when the ratio of CM dose (mL) to creatinine clearance (mL/min) was >6.15 [38].

At the same time, there are numerous reports that cast doubt on the important role of CM as a major etiologic factor for AKI in patients with AMI undergoing PCI. In one study, a comparison of a group of STEMI patients who received CM at the time of PCI with a group treated with fibrinolytic or conservative therapy showed no significant difference in the incidence of AKI. The occurrence of AKI was associated with older age, reduced baseline GFR, heart failure and hemodynamic instability [39]. Another study examined the effect of repeated coronary interventions on renal function in a group of 2942 patients who underwent 1 to 9 procedures. In patients with multiple CM exposures, deterioration of renal function over time was associated with known risk factors for renal disease progression, but not with cumulative CM volume [40]. In a study involving patients with AMI and MVCAD presenting with cardiogenic shock, rates of AKI were not significantly different between the total revascularization and ischemic lesion-only revascularization groups [41]. Moreover, metanalyses performed did not show an increased risk of CI-AKI in patients with STEMI undergoing multivessel revascularization during a single hospitalization compared with revascularization of the culprit lesion only [42]. Mukete et al. in their metanalysis found a slight trend toward a reduction in CI-AKI despite the larger volume of CM used in patients with STEMI undergoing multivessel revascularization compared to patients who underwent revascularization of the ischemic lesion only. Moreover, this metanalysis showed a significantly reduced rate of cardiovascular death and repeat revascularization among patients undergoing multivessel revascularization [43].

Among the available publications, there are also those that question the important role of CM as a causative factor for renal failure in critically ill patients hospitalized for other causes [44]. McDonald et al. in their study reported a similar incidence of AKI among patients who underwent CT scans with and without CM administration. Moreover, administration of CM was not associated with higher 30-day mortality or the need for dialysis. The above results may be explained by the presence of AKI risk factors other than CM administration in severely ill patients, whose role in the development of AKI may be overestimated [45,46]. In an analysis of six million hospitalizations, no differences were found in the rate of AKI between patients who received and did not receive CM. The frequency of AKI in both groups increased with the number of comorbidities. According to the authors, the frequency of AKI associated with CM may be overstated in the literature and overestimated by clinicians [47]. A retrospective analysis of a large group of hospitalized patients who did not receive CM led to similar conclusions. In this group, creatinine levels increased to values at which AKI was diagnosed as often as reported in groups of patients who received CM [48].

## 3. Risk Factors for Acute Kidney Injury in Patients with Myocardial Infarction

### 3.1. Severity of Infarction and Heart Failure

All papers evaluating risk factors for renal damage in patients with AMI list features related to infarct severity and hemodynamic instability as independent predictors of AKI. Patients with symptoms of heart failure/cardiogenic shock, lower ejection fraction and larger left ventricular end-diastolic dimension, high levels of markers of myocardial damage, longer-lasting symptoms and extensive anterior wall infarction had a significantly higher risk of AKI during hospitalization [15,16,17,18,21,49,50]. After analyzing data from nearly one million patients from the National Cardiovascular Data Registry Cath-PCI registry, three of the strongest predictors of AKI in patients undergoing PCI were identified. Two of these factors were specifically related to AMI severity and hemodynamic status—a diagnosis of STEMI and cardiogenic shock; the third factor was advanced chronic kidney disease. The association between the occurrence of AKI and the volume of CM used was much less significant, than it was expected [19]. Acute heart failure is one of the most common complications of AMI. Renal damage due to heart failure is referred to as type I cardiorenal syndrome. A decrease in cardiac output leads to decreased vascular bed filling and renal hypoperfusion, an increase in renal vein pressure and a decrease in GFR. In addition, hemodynamic changes cause activation of the sympathetic nervous system and the RAAS, which exacerbates vasoconstriction and further renal damage [51].

### 3.2. Comorbidities and Others Factors

In available analyses, the incidence of AKI was higher in older patients [15,17,21,49]. In one study, female gender was associated with a higher risk of AKI [21]. Multimorbidity also significantly increases the risk of developing AKI in AMI. In clinical studies, patients with impaired baseline renal function, a history of diabetes, hypertension, hyperuricemia or anemia were more likely to develop AKI [52,53].

### 3.3. Malnutrition

Malnutrition has been a frequently discussed topic in recent years in the context of adverse clinical outcomes in cardiovascular disease. This factor deserves special attention, including in terms of the risk of AKI in patients with AMI. There are data supporting an association between nutritional status and the risk of AKI. A retrospective, multicenter study involving more than three thousand patients with acute coronary syndrome (ACS) showed that assessment of malnutrition risk using the Nutritional Risk Screening 2002 (NRS-2002) scale was useful in identifying patients at high risk for AKI. The group of patients with an NRS-2002 score ≥ 3 had a higher incidence of AKI and higher mortality from any cause [54]. In another study, the prognostic nutrition index (PNI), calculated from serum albumin levels and total lymphocyte count, correlated with the risk of AKI in STEMI patients undergoing PCI [55]. Low plasma albumin levels alone have also been shown to be an independent predictor of AKI in ACS patients undergoing PCI [56]. Based on the available data, it appears that malnutrition status may be a risk factor for AKI in severely ill patients, regardless of their use of CM. Among elderly patients hospitalized in the intensive care unit, the group diagnosed with AKI had a higher risk of malnutrition as assessed by modified Nutrition Risk in Critically Ill (mNUTRIC) compared to the group without AKI [57]. Further studies may reveal whether the risk of AKI in patients with malnutrition and MVCAD differs depending on the revascularization strategy adopted and the amount of CM used. Correcting malnutrition and hypoalbuminemia in terms of reducing the incidence of CI-AKI is also a controversial issue. Knowing that reduced albumin levels may be the cause of “intravascular hypovolemia”, one could speculate that correcting protein deficiencies could benefit renal function. To date, no such studies have been conducted, and it seems crucial in this situation to identify and recognize malnutrition in patients undergoing PCI.

### 3.4. Hydration Status

Adequate hydration status contributes to maintaining adequate cardiac output, increasing blood volume and improving renal perfusion. Dilation of the intravascular space is thought to reduce renal vasoconstriction through inhibition of vasopressin secretion, inhibition of the RAAS and increased synthesis of vasodilators. In addition, adequate hydration reduces the concentration of CM in the renal tubules alleviating direct damage to renal tubular epithelial cells. On the other hand, excessive fluid loading may worsen the prognosis, especially in patients with left ventricular dysfunction. Methods of optimizing hydration with proven efficacy are discussed in the section of the article on ways to prevent AKI.

### 3.5. Medications

The use of loop diuretics has been an independent risk factor for AKI in patients with AMI [17,18,49]. Drugs from this group cause activation of the sympathetic nervous system and the RAAS, resulting in increased peripheral vascular resistance and decreased renal perfusion contributing to the development of AKI [58]. An association has also been found between the use of RAAS inhibitors and a higher rate of AKI development [15]. This fact can be explained by excessive lowering of blood pressure in some patients, and thus reduced renal perfusion.

### 3.6. Vascular Access

Most studies comparing the effect of the type of vascular access on the rate of AKI in patients undergoing PCI have shown a lower risk of AKI in the transradial access group [59,60]. Transradial access is associated with a lower risk of major bleeding associated with vascular access. Bleeding is a significant risk factor for AKI [61]. Thus, transradial access may reduce episodes of AKI by reducing the risk of hemodynamic instability associated with bleeding. A second mechanism that reduces the risk of AKI with transradial access is reduced cholesterol embolization in the renal arteries. Through the transradial access an injury to the descending aorta from catheter passage is avoided [62]. It is noteworthy, however, that the SAFARI-STEMI trial produced results that undermine the independent association between vascular access and the risk of AKI noted in earlier studies. In STEMI patients included in the study, there was no association between the site of vascular access and AKI [63].

## 4. New Markers of Kidney Damage

Research is underway to identify an effective marker for early identification of patients with a high risk of AKI. Plasma calprotectin and neutrophil gelatinase-associated lipocalin (NGAL) have been shown to be significantly higher in patients with AKI [64]. Tung et al. found an association between high levels of brain natriuretic peptide (BNP) and soluble suppression of tumorigenicity-2 (sST2), as well as biomarkers of renal injury, i.e., NGAL and cystatin C, and the risk of AKI in STEMI patients undergoing PCI. According to the authors of that study, the combined evaluation of the above biomarkers can help stratify the risk of AKI in patients with STEMI [65]. The role of cystatin C as a potential alternative to serum creatinine in patients with low muscle mass should be noticed. Serum creatinine concentration is known to be dependent on the balance between creatinine production by the muscles and the elimination by the kidney. Patients with cardiovascular diseases are often characterized by low muscle mass, and, therefore, the serum creatinine concentration and the GFR obtained from serum creatinine do not reflect the actual kidney function in this group. Cystatin C is a marker unaffected by potential determinants such as muscle mass and diet. Recent studies indicate that the calculation of GFR based on cystatin C concentration may be a supplement or alternative to methods based on creatinine concentration [66]. Urinary insulin-like growth factor-binding protein 7 (IGFBP7) and tissue metalloproteinase inhibitor-2 (TIMP-2) are urinary G1 cell cycle arrest biomarkers. In the Sapphire study, they were superior to the other tested biomarkers in predicting stage 2 or 3 of the AKI [67]. In another study, an early increase in urinary G1 cell cycle arrest biomarkers after transaortic aortic valve implantation (TAVI) predicted postoperative AKI with high accuracy [68]. Urinary L-type fatty acid-binding protein (L-FABP) is released from the epithelial cells of the proximal tubules into the urine in response to kidney damage. Data suggest that urinary L-FABP in combination with other renal markers as NGAL and kidney injury molecule-1 (KIM-1) effectively detect AKI at an early stage [69]. There are also data indicating that quantitative assessment of endothelial activity based on plasma thrombomodulin (TM) and angiopoietin (Ang-2) levels may help identify patients with AMI complicated by renal damage at an early stage [70]. Interleukin-18 (Il-18) or interferon-gamma inducing factor belongs to the interleukin-1 family and is formed after cleavage by caspase. This cytokine is produced by the intercalated cells of the proximal tubule and is believed to be an AKI marker associated with ischemia. While animal models have shown promising results regarding the predictive value of Il-18 in predicting AKI, the data from human studies are inconclusive. An analysis of 11 studies showed a moderate diagnostic value of IL-18 [71]. Other inflammatory markers such as the systemic immune-inflammatory response index (SII) and high-sensitivity C-reactive protein (hsCRP) may also prove useful [72,73]. These new biomarkers can aid risk stratification, early detection and early intervention in case of AKI in patients with AMI. However, further prospective studies are needed to validate their predictive value.

## 5. Ways of Prevention

### 5.1. Hydration

Hydration is, for the time being, the only method of preventing and treating CI-AKI with fully proven efficacy. Intravenous hydration appears to have advantages over oral hydration due to more precise dosing and better control. A meta-analysis by Wang et al. showed that in STEMI patients undergoing PCI, perioperative hydration with isotonic saline solution was associated with a significant reduction in the rate of CI-AKI (16.9% in the hydration group versus 26.4% in the control group) [74]. Isotonic hydration (0.9% sodium chloride) appeared to be more effective than hypotonic hydration (0.45% sodium chloride plus 5% glucose) in preventing CI-AKI [75]. The use of sodium bicarbonate remains controversial. A meta-analysis by Jang et al. demonstrated the superiority of sodium bicarbonate-based hydration over sodium chloride in preventing CI-AKI in patients exposed to iodinated CM. However, other studies have not confirmed this advantage [76,77]. Despite the undoubted efficacy of hydration in the prevention of CI-AKI, however, excessive fluid supply should be avoided, especially in patients with left ventricular dysfunction and advanced renal failure. Data suggest that higher hydration volume administered at standard times is not associated with a reduced risk of CI-AKI, nor with a better long-term prognosis. Excessive hydration volume ratios (hydration volume/body weight; HV/BW) in patients with impaired renal function, may even be associated with an increased risk of CI-AKI and death [78]. According to the currently accepted regimen, it is recommended to use isotonic saline at a dose of 1 mL/kg b.w./h for 12 h before the administration of CM and continue hydration at the same dose 24 h after the procedure. For patients with heart failure (LVEF ≤ 35% or symptoms > NYHA class II), the dose should be reduced to 0.5 mL/kg b.w./h [79]. There are also additional methods of optimizing hydration with proven efficacy in the prevention of CI-AKI. The use of an automated system that adjusts the volume of fluid administered intravenously to the amount of urine excreted (UFR, urine flow rate) is a non-invasive method of optimizing hydration, whose effectiveness in reducing CI-AKI has been confirmed in randomized trials [80]. Another non-invasive method proposed to determine the appropriate volume of hydration fluids is bioelectrical impedance vector analysis (BIVA). In The HYDRA study, patients with low BIVA (i.e., dehydrated) on admission undergoing coronary angiography were randomly assigned to a standard or double volume saline hydration group. The incidence of CI-AKI was significantly lower in the dual hydration group compared to the standard group (11.5% vs. 22.3%) [81]. Optimization of hydration in the prevention of AKI can also be carried out by invasive methods. Among patients with congestive heart failure and chronic kidney disease, a randomized trial showed a reduction in the incidence of CI-AKI in the central venous pressure (CVP)-guided hydration group compared with the control group (15.9% vs. 29.5%). The incidence of acute heart failure did not differ between the two groups [82]. In the Prevention of Contrast Renal Injury with Different Hydration Strategies (POSEIDON) trial, patients with preexisting renal failure and an additional risk factor were assigned to the standard or LVEDP-controlled hydration group. A significant reduction in CI-AKI was found in the left ventricular end-diastolic pressure (LVEDP)-controlled hydration group compared to the standard group [83]. The REMEDIAL III trial comparing UFR- and LVEDP-controlled hydration in terms of the incidence of CI-AKI and acute pulmonary edema is currently underway.

### 5.2. Medications

A number of drugs have been tested for efficacy in the prevention of CI-AKI. Rosuvastatin in moderate-to-high doses has been shown to have a beneficial effect in reducing the incidence of CI-AKI in patients undergoing PCI [84,85,86]. In the past, there were opinions that N-acetylcysteine significantly reduces the incidence of CI-AKI. Unfortunately, randomized trials have failed to demonstrate the benefit of oral N-acetylcysteine in the prevention of CI-AKI in patients with chronic kidney disease and high risk of CI-AKI undergoing angioplasty [87,88]. Diuretics also failed in a retrospective study involving 1061 patients, as it was shown that their use may even increase the risk of CI-AKI after PCI. In patients at risk of CI-AKI, their use should be used with caution to avoid exacerbating renal damage due to reduced intravascular volume [89]. Previous studies on the effect of RAAS inhibitors on the risk of AKI have not provided a clear answer. In the CAPITAN trial involving 208 patients with moderate renal failure undergoing cardiac catheterization, withholding angiotensin-converting-enzyme inhibitors (ACEI) or angiotensin receptor blockers (ARB) resulted in a nonsignificant reduction in the risk of CI-AKI and a significant reduction in the post-procedure increase in creatinine [90].

### 5.3. Hemodialysis/Hemofiltration

Most of the clinical trials conducted found no benefit from immediate hemodialysis after CM exposure in patients with existing chronic kidney disease undergoing angiography [91]. However, Marenzi et al. found that hemofiltration performed before and after CM exposure may be a strategy for preventing CI-AKI in patients with pre-existing chronic kidney disease, also reducing the need to implement renal replacement therapy as well as mortality [92,93].

## 6. Conclusions

Acute kidney injury is a common (AKI) and potentially dangerous complication of acute myocardial infarction (AMI) with a complex etiology. (Figure 2). Factors related to the severity of the infarction and the patient’s hemodynamic status, as well as individual factors, primarily the presence of advanced chronic kidney disease, appear to play a major role in its pathogenesis. The correlation between the volume of contrast media (CM), the interval between its doses, and the risk of AKI seems much less expressed and not fully understood. In the face of objective findings, the risk of AKI closely associated with CM in patients undergoing multivessel percutaneous coronary intervention (PCI) seems overestimated and may be an unwarranted reason for postponing revascularization of non-culprit lesions. Further studies are needed to determine the optimal timing of multivessel revascularization in patients with AMI. These studies should include assessment of the risk of AKI, its control, detection of early-phase AKI, including based on new biochemical markers, and optimal preventive and therapeutic management strategies. Reducing the episodes of AKI may have important implications for the prognosis of patients with AMI and multivessel coronary artery disease.

## Figures and Tables

**Figure 1 nutrients-15-00021-f001:**
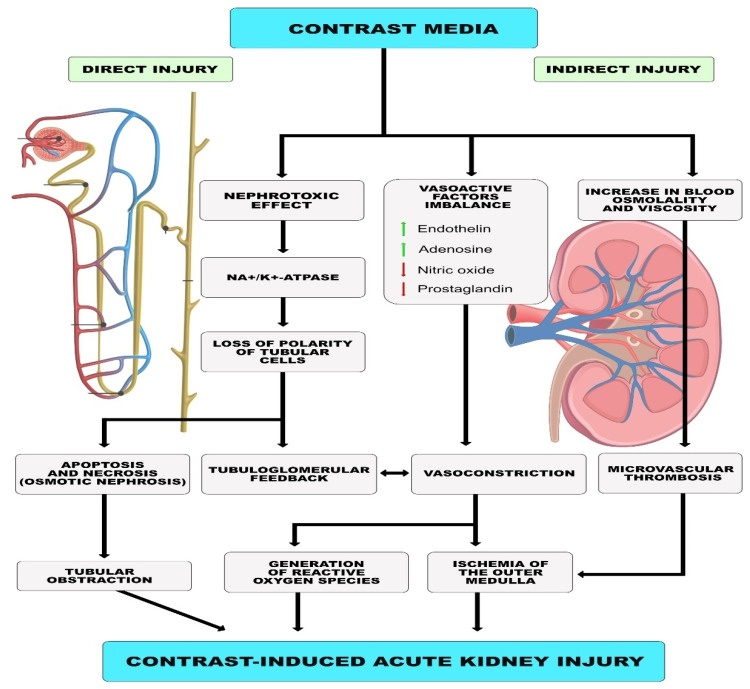
Proposed mechanism of contrast-induced acute kidney injury.

**Figure 2 nutrients-15-00021-f002:**
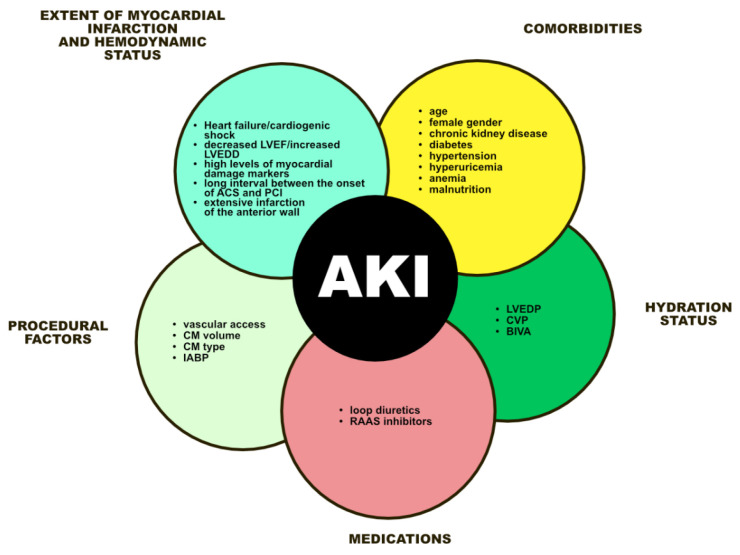
Diagram illustrating risk factors for AKI in patients with MI undergoing PCI AKI, acute kidney injury; LVEF, left ventricle ejection fraction; LVEDD, left ventricular end-diastolic dimension; PCI, percutaneous coronary intervention; CM, contrast media; IABP, intra-aortic balloon pump; RAAS, renin-angiotensin-aldosterone system; LVEDP, left ventricular end-diastolic pressure; CVP, central venous pressure; BIVA, bioelectrical impedance vector analysis.

**Table 1 nutrients-15-00021-t001:** Diagnostic criteria for AKI according to RIFLE, AKIN and KDIGO. RIFLE, Risk Injury Failure Loss and End-Stage Renal Failure; AKIN, Acute Kidney Injury Network; KDIGO, Kidney Disease: Improving Global Outcomes; Scr, serum creatinine; RRT, renal replacement therapy.

**Class/Stage**	**Criterion SCr/GFR**			**Urine Output**
	**RIFLE**	**AKIN**	**KDIGO**	**RIFLE/AKIN/KDIGO**
Risk (RIFLE)/ Stage 1 (AKIN and KDIGO)	SCr increase ×1.5 within 7 days or GFR decrease >25%	SCr increase ≥26.5 μmol/L (≥0.3 mg/dL) within 48 h or 1.5–2× within 7 days	SCr increase ≥26.5 μmol/L (≥0.3 mg/dL) within 48 h or 1.5–1.9× within 7 days	<0.5 mL/kg/h (>6 h)
Injury (RIFLE)/ Stage 2 (AKIN and KDIGO)	SCr increase ×2 or GFR decrease >50%	SCr increase 2–3×	SCr increase 2–2.9×	<0.5 mL/kg/h (>12 h)
Failure (RIFLE)/ Stage 3 (AKIN and KDIGO)	SCr increase ×3 or SCr ≥ 354 μmol/L (≥4 mg/dL) with acute rise 44 μmol/L (>0.5 mg/dL) or GFR decrease >75%	SCr increase ×3 or SCr ≥ 354 μmol/L (≥4 mg/dL) with acute rise 44 μmol/L (>0.5 mg/dL) or need for RRT	SCr increase ×3 or SCr ≥ 354 μmol/L (≥4 mg/dL) or need for RRT	<0.3 mL/kg/h (>24 h) or anuria >12 h

## Data Availability

Not applicable.

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
