# Peer review of "What Promotes Acute Kidney Injury in Patients with Myocardial Infarction and Multivessel Coronary Artery Disease—Contrast Media, Hydration Status or Something Else?"

_nutrients, 2022, doi:10.3390/nu15010021_

Round 1

Reviewer 1 Report

Basically, this review appears extensive, reliable and informative.

Only one topic needs some supplementary studies: new markers of kidney damage. In my opinion, some other markers that appear interesting and disposable such as TIMP-2 and Il-18 should be discussed.

Author Response

Dear Reviewer

Thank you for a thoughtful and helpful review of our paper. The comments have helped us improve the manuscript. Following your suggestions, we have made revison of the manuscript.

1. Comment

In my opinion, some other markers that appear interesting and disposable such as TIMP-2 and Il-18 should be discussed.

Response

We have extended the chapter on new biomarkers, and we have discussed the role of: Il-18, TIMP-2, IGFBP7, L-FABP.

We look forward to your response and hope the revisions will enable you to accept this version of the manuscript.

Authors

Reviewer 2 Report

The paper is interesting and concerns the current clinical problem. I have some minor remarks.

There are some international  and national recommendations which should be cited in this review. The most important are KDIGO but also Italian are well written:

https://kdigo.org/wp-content/uploads/2016/10/KDIGO-2012-AKI-Guideline-English.pdf

Orlacchio A, Guastoni C, Beretta GD, et al. SIRM-SIN-AIOM: appropriateness criteria for evaluation and prevention of renal damage in the patient undergoing contrast medium examinations-consensus statements from Italian College of Radiology (SIRM), Italian College of Nephrology (SIN) and Italian Association of Medical Oncology (AIOM). Radiol Med. 2022;127(5):534-542. doi:10.1007/s11547-022-01483-8

Please enrich the article with these recommendations in the relevant chapters.  

Chapter 2.3

The risk factors which are associated with contrast induced nephropathy should be enumerated.

Chapter 4

Many patients with CVD suffer from muscle wasting therefore serum creatinine level did not show precisely glomerular filtration. The role of serum cystatin C level as a marker independent from muscle mass and useful in estimation of GFR should be discussed.  

Some detailed remarks and suggestions :

Figure 1.

„Proposed” is not necessary „ Mechanism of contrast-induced acute kidney injury” is sufficient

Line 202 : “much less “ are not the most suitable words

Line 209 “ Individual factors “ should be changed for “other factors” or “comorbidities”

Line 368 “high hopes “ – it is not an appropriate term

Author Response

Dear Reviewer

Thank you for a thoughtful and helpful review of our paper. The comments have helped us improve the manuscript. Following your suggestions, we have made revison of the manuscript.

1. Comment

There are some international and national recommendations which should be cited in this review.

Response: We agree with the comment. We have cited: 2012 KDIGO guidelines and recent statements from the Italian College of Radiology (SIRM), the Italian College of Nephrology (SIN) and the Italian Association of Medical Oncology (AIOM) in the indicated chapters. 2. Comment

Some detailed remarks and suggestions

Figure 1.

Proposed” is not necessary „ Mechanism of contrast-induced acute kidney injury” is sufficient

Line 202 : “much less “ are not the most suitable words

Line 209 “ Individual factors “ should be changed for “other factors” or “comorbidities”

Line 368 “high hopes “ – it is not an appropriate term

Response: Revised accordingly.

We look forward to your response and hope the revisions will enable you to accept this version of the manuscript.

Authors
